# Leveraging Gene Redundancy to Find New Histone Drivers in Cancer

**DOI:** 10.3390/cancers15133437

**Published:** 2023-06-30

**Authors:** Daria Ostroverkhova, Daniel Espiritu, Maria J. Aristizabal, Anna R. Panchenko

**Affiliations:** 1Department of Pathology and Molecular Medicine, Queen’s University, Kingston, ON K7L 3N6, Canada; 20do7@queensu.ca (D.O.); 15dise@queensu.ca (D.E.); 2Department of Biology, Queen’s University, Kingston, ON K7L 3N6, Canada; 3Department of Biomedical and Molecular Sciences, Queen’s University, Kingston, ON K7L 3N6, Canada; 4School of Computing, Queen’s University, Kingston, ON K7L 3N6, Canada; 5Ontario Institute of Cancer Research, Toronto, ON M5G 0A3, Canada

**Keywords:** histones, nucleosome, gene redundancy, cancer driver gene, cancer mutation, computational method, mutational signatures, mutational processes, chromatin

## Abstract

**Simple Summary:**

Histones are a group of proteins that are essential for chromatin function. Histone coding genes have been found to be mutated in a number of cancers but how these mutations arise and whether they contribute to cancer biology remains largely unknown. Answering these questions is difficult because histone proteins are encoded by large families of genes that are thought to be redundant. Here, we overcome these limitations using a large dataset of whole exome sequencing data from cancer patients representing 68 different cancer types. We find that specific cancer types exhibit disproportionate susceptibilities to accruing mutations in histone genes versus other genes. We delineate factors influencing the probability of mutations accumulation in histone genes and detect new histone gene drivers.

**Abstract:**

Histones play a critical role in chromatin function but are susceptible to mutagenesis. In fact, numerous mutations have been observed in several cancer types, and a few of them have been associated with carcinogenesis. Histones are peculiar, as they are encoded by a large number of genes, and the majority of them are clustered in three regions of the human genome. In addition, their replication and expression are tightly regulated in a cell. Understanding the etiology of cancer mutations in histone genes is impeded by their functional and sequence redundancy, their unusual genomic organization, and the necessity to be rapidly produced during cell division. Here, we collected a large data set of histone gene mutations in cancer and used it to investigate their distribution over 96 human histone genes and 68 different cancer types. This analysis allowed us to delineate the factors influencing the probability of mutation accumulation in histone genes and to detect new histone gene drivers. Although no significant difference in observed mutation rates between different histone types was detected for the majority of cancer types, several cancers demonstrated an excess or depletion of mutations in histone genes. As a consequence, we identified seven new histone genes as potential cancer-specific drivers. Interestingly, mutations were found to be distributed unevenly in several histone genes encoding the same protein, pointing to different factors at play, which are specific to histone function and genomic organization. Our study also elucidated mutational processes operating in genomic regions harboring histone genes, highlighting *POLE* as a factor of potential interest.

## 1. Introduction

Histones are evolutionarily conserved intranuclear proteins that uphold the integrity and function of eukaryotic genomes. Histones function primarily within the context of nucleoprotein complexes known as nucleosomes and chromatosomes (Figure 1A). The nucleosome consists of ~147 base pairs of DNA wrapped around an octamer comprised of two proteins from each of the H2A, H2B, H3, and H4 histone families. A chromatosome is an extension of the nucleosome and consists of a nucleosome, an H1 linker-histone, and “linker” DNA. Interactions between neighboring nucleosomes or chromatosomes allow for the formation of chromatin structure. Through facilitating interactions between DNA and other intranuclear proteins, histones participate in processes of DNA replication and repair, gene expression, and chromatin compaction. Moreover, as key elements of the epigenome, histones function in mitotic memory, preserve cellular identity and lineage fidelity, and mediate chromatin remodeling during embryonic development.

Although critical for chromatin stability and function, histone genes are susceptible to mutagenesis. In recent years, numerous histone mutations have been observed in several different cancers, and a handful of these mutations have been demonstrated to associate with distinct cellular phenotypes and patient prognoses [2,3,4,5,6]. Studies examining the prevalence of histone mutations in a wide range of cancers have estimated that histone mutations occur in approximately 4% of all cancer cases [2]. At the genesis of each mutation, there is a triggering event that leads to a genetic alteration. This can be driven by extrinsic factors such as tobacco smoke, or internal factors such as aberrant APOBEC/AID cytidine deaminase [7,8] or DNA Polymerase POLE/POLD1 activities [9,10]. The subsequent failure of a cell to repair such mutations can lead to their fixation in the genome and propagation to progeny unless a substantial selective disadvantage is conferred. Both stochastic and deterministic factors influence the fixation of a mutation in cancer cell populations. These include the local DNA sequence and genomic location; the cancer, tissue, and cell type; the anatomical location; the external mutagenic exposures; and the effect of the mutation on cell fitness [11,12]. To understand the etiology of histone mutations, we must understand how the factors described above promote the fixation of histone mutations in the genome.

Histones are peculiar in that they are encoded by multiple genes. For each of the five histone families, many genes encode identical or highly similar histone proteins that can be used interchangeably in the nucleosome/chromatosome and in most cases are thought to perform similar functions. The overwhelming majority of histone genes are clustered in three genomic regions: 60 histone genes are found in a cluster on chromosome 6, 10–12 genes on a cluster on chromosome 1, and four genes on another chromosome 1 locus [13]. Clustered histone genes correspond to canonical, replication-dependent histones, which do not contain introns and undergo unique 3′ end processing during transcription [14]. As the name suggests, the transcription of canonical histones is tightly coupled to DNA replication, with over- and under-expression being toxic to cells [15,16].

In contrast to canonical histones, variant histones differ from their canonical counterparts in genomic localization, expression, and function. In comparison to canonical histones, variant histone genes do not form clusters in the genome and are more dispersed. Moreover, their mRNA products can be expressed at any point in the cell cycle, their transcripts are polyadenylated, and they contain introns, which means that they can also produce multiple transcripts and different protein isoforms.

Several reasons may underscore the apparent redundancy of histone genes. On one hand, the importance of histone function may warrant high gene redundancy to protect cells against loss-of-function mutations. In this case, a large mutational load on histone genes may result from most mutations being inert or having limited impacts on overall cellular fitness. The redundancy of histone genes may also provide the cell with the ability to rapidly produce histone proteins during DNA replication. Specifically, it is estimated that mammalian cells must produce ~400 million histone proteins during the S phase, a stage of the cell cycle that spans just a few hours [17].

The biology of histone genes complicates our ability to understand their roles in cancer. In addition, while some works describe the effects of histone mutations in cancer, literature focusing on the etiology of these mutations is sparse. Here, armed with a large data set of cancer mutational data collected from various studies and encompassing 68 cancer types, we describe mechanisms of mutation occurrence in histone genes and discover new histone driver genes for each cancer type. Our study paves the road for developing future computational methods that better predict the functional consequence of mutations on histone genes.

## 2. Materials and Methods

### 2.1. Mutation Data Collection and Processing

Publicly available whole exome sequencing (WES) and clinical data were gathered from the Genomic Data Commons (GDC) for 12,989 cancer patients within The Cancer Genome Atlas, Multiple Myeloma Research Foundation CoMMpass, Human Cancer Model Initiative, Count Me In (CMI), and the Clinical Proteomic Tumor Analysis Consortium (CPTAC) programs [18]. Patients were grouped into 68 specific cancer types based on their primary diagnoses. Primary diagnoses followed morphological classifications from the third edition of the International Classification of Diseases for Oncology. For statistical analyses, only cancer types with at least 10 patients with the same primary diagnosis were considered. An additional pan-cancer model was created, where all patients were grouped together. To identify somatic histone mutations within the WES data, a list of histone gene names was compiled from the Human Genome Organization Nomenclature Committee (HGNC) and Histone DataBase 2.0 [19,20]. This list of histone genes includes a total of 96 currently approved histone genes with 59 encoded histone proteins (Appendix A).

### 2.2. Comparisons of the Observed Mutation Rates at the Protein Level

Full-length coding sequences were gathered for 19,084 human genes from the Consensus Coding Sequence (CCDS) database [21]. From the total number, missense, silent, nonsense, nonstop, frameshift insertion, frameshift deletion, splice site, splice region, and translational start site mutations were counted for each gene detailed in the CCDS data (Appendix A). Genes were grouped together if they encoded a common protein. This was done because histone genes are unusual in that multiple genes encode the same protein. Therefore, the average observed mutation rate per base pair for a given protein was calculated as follows:$$r\;=\;\frac{\sum_{i=1}^{n}m_{i}}{\sum_{i=1}^{n}c_{i}}$$
where *n* is the total number of genes encoding a given protein, *m_i_* is the total number of mutations within gene *i*, and *c_i_* is the coding sequence length of gene *i* in base pairs. Mutation rate estimates were calculated for each protein in each cancer type and for the pan-cancer model (Appendix A).

### 2.3. Mutation Rate Comparison between Histones and Known Cancer Drivers at the Protein Level

Genes known to drive cancer development were gathered from the Catalog of Somatic Mutations in Cancer (COSMIC) Cancer Gene Census (CGC) [22]. We removed histone genes from the CGC to ensure that we were comparing mutation rates in histones to known non-histone cancer drivers. For COSMIC CGC genes, we only focused on Tier 1 genes (578 genes), those with the strongest evidence of playing a role in the development of specific cancer types. Uniprot identifiers for CGC driver gene protein products were gathered using the Uniprot batch retrieval tool, returning 575 CGC driver proteins [23]. For each of our cancer types, we compared the mutation rates of all Tier 1 CGC driver proteins to those of histones. Although CGC driver genes have been annotated for individual cancer types, this information could not be considered in the analysis because our cancer type classification is different from the one used in the CGC data sets.

### 2.4. Mutation Rate Variation across Histone Genes Encoding Common Proteins

Chi-Squared tests were performed for each cancer type to determine if the frequency of mutations differs across histone genes encoding a common protein. For this analysis, we considered missense, silent, nonsense, nonstop, frameshift insertion, frameshift deletion, splice site, splice region, and translational start site mutations and counted the number of patients with a mutation in each gene. Statistical tests were performed if at least one histone gene was mutated in a given cancer type. To increase the robustness of our analysis for sparse data sets, *p*-values were obtained using Monte Carlo simulations, instead of Chi-Squared tests (10,000 steps) when the expected values from a given contingency table were less than 5 observations [24].

### 2.5. Identification of New Driver Histone Genes

The state-of-the-art MutSig2CV algorithm was used to identify novel driver histone genes in each specific cancer type [25,26]. Briefly, MutSig2CV performs an analysis of somatic mutations to identify genes mutated more frequently than expected by chance relative to a background mutation rate model. The background mutation rate is not constant across the genome and varies as a result of genomic and epigenomic factors. To build a background mutation rate model, the algorithm takes into account observed mutation rate, tri-nucleotide context-specific mutability, and gene-specific covariates including gene expression levels, DNA replication time, and chromatin state (open/closed). We defined genes to be significantly mutated (driver genes) if MutSig2CV reported a q-value ≤ 0.1. In order to examine the relationship between mutation rate and large-scale confounding factors such as replication time, chromatin state, and expression level, we used the covariates provided by MutSigCV. The MutSigCV covariates, in turn, were taken from the following experimental studies. The gene expression data were taken from the Cancer Cell Line Encyclopedia [27], where average expression level across 91 cell lines was reported. Replication time [28] was measured in HeLa cells, and its values ranged from 100 (very early) to 1500 (very late). Chromatin state data were taken from the experimental Hi-C data generated in the karyotypically normal human lymphoblastoma cell line (GM06990) and aberrant erythroleukemia cell line (K562) [29]. Hi-C maps physical contacts on genomic DNA at one megabase or higher resolution. MutSigCV only uses Hi-C data from the K562 cell line. Values for the chromatin state range from −50 (very closed) to +50 (very open). MutSigCV covariates are available in Appendix A [25]. Chromosome locations for histone genes were obtained for the human reference genome (GRCh38) in the ENCODE data portal [30]. Variant and canonical histone gene information was gathered from previous reports [5,13,31].

### 2.6. Mutational Profiles and Mutational Signature Detection

To identify which mutational processes shape the landscape of cancer histone mutations and to determine if mutational processes differ between histone and non-histone genes, all patients were divided into two sets: “patient_set1” contained all patients with at least one histone mutation, while “patient_set2” contained patients without a histone gene mutation. In “patient_set1”, mutations were further divided into those occurring on histone (referred to as “gene_group1”) and non-histone genes (“gene_group2”). All genes mutated in patient_set2 belonged to “gene_group3” (Figure 2A).

The footprint of mutational processes operating in a given cancer type or a group of genes can be represented as a probability mass function of a multinomial distribution over 96 trinucleotides. We constructed a 96-dimensional profile vector for each gene group. Next, we decomposed the mutational profile of each gene group into a set of mutational signatures using the MutaGene “identify” module [32]. This module employs a maximum likelihood algorithm, which can also account for context-free mutational processes. We used this module because it allows for the accurate detection of true mutational processes, including processes that do not depend on the local sequence context. The GRCh38 human genome was used as the reference genome assembly. To assess the robustness of profile decomposition into mutational signatures, a bootstrap method was used. Each profile was resampled 100 times, with replacement by drawing the same number of mutations that were used to construct a given profile. Each bootstrap profile was then decomposed into signatures. From the resulting distributions, the mean values of exposure and 95% confidence intervals were calculated. A set of 67 single-based substitution (SBS) signatures from the Catalogue of Somatic Mutations in Cancer (COSMICv3) [33] was utilized in the study. A signature was determined to be active if its exposure was higher than 5%. Exposure refers to the percentage of query mutations explained by a given mutational process. We merged mutational signatures with a common annotated etiology: UV-light related signatures (SBS 7a–d), DNA mismatch repair (MMR)-linked signatures (SBS 6, 15, 21, 26, 44); POLE-deficient (10a–b); POLD1-deficient (SBS 10c–d); Tobacco smoking–associated signatures (SBS 4, 92); APOBEC (SBS 2,13).

## 3. Results

### 3.1. Histone Mutation Rates Vary across Cancer Types

First, we asked if histone mutation rates varied between cancer types and found that the mutation rate of histone genes differed significantly between the 68 cancer types present in our data set (Kruskal–Wallis and post hoc Dunn test, *p*-value << 0.001). In this analysis, we also compared the mutation rate of histone and non-histone proteins in all cancer patients (pan-cancer). Overall, we found significantly higher mutation rates in histones compared to non-histone proteins in pan-cancer and seven other cancer types (Mann–Whitney U, q-value < 0.05, Benjamini–Hochberg, FDR = 0.05, Figure 1B—blue points). The most pronounced differences were observed for malignant lymphoma, translational cell carcinoma, and squamous cell carcinoma. In contrast, five cancer types had histone mutation rates that were significantly lower when compared to non-histone proteins, although we note that the effect size of the difference was small. No significant difference between the mutation rate of histone proteins and non-histone proteins was found in any other cancer type (Appendix A).

### 3.2. Comparison of Histone Mutation Rates to List of Cancer Drivers

The significantly increased mutation rate observed for histones in some cancer types prompted us to examine if the mutation rate of histone proteins is different than that of non-histone proteins known to drive cancer development (“CGC driver proteins”, see Section 2). A focus on histone proteins is warranted because several histone genes code for the same protein. The CGC driver protein list contains three histones (H3.1, H3.3, and H4), which were removed from the CGC data set and grouped with the histone proteins for this analysis. For seven cancer types, we found that histone proteins had a significantly higher mutation rate compared to non-histone CGC driver proteins (Mann–Whitney U, q < 0.05, Benjamini–Hochberg, FDR = 0.05, Appendix A). An opposite effect was observed for 17 cancer types (complete results can be found in Appendix A), effects that were similar to the results described above, when histone genes were considered individually.

### 3.3. Observed Mutation Rates Do Not Vary across Histone Families

A large proportion of the literature on histone cancer driver mutations has focused on H3 and H1 histones [34,35,36,37]. However, mutations in other histone families have been highlighted as potential drivers of oncogenesis. To determine if mutation rates vary between histone families (H1, H2A, H2B, H3, H4) Kruskal–Wallis tests were performed. For this analysis, only cancer types that had at least 10 mutations in each of the histone types were considered. No statistically significant difference between histone family mutation rates was found for any of the 12 cancer types considered, including the pan-cancer model (q ≥ 0.05, Benjamini–Hochberg, FDR = 0.05). These findings are surprising given the current focus on the H3 and H1 family of histones in cancer biology. Our results suggests that a larger scope of histone families may contribute to cancer biology.

### 3.4. Genes Encoding the Same Histone Protein Differ in Mutation Frequencies

Given that several histone proteins are encoded by more than one gene, we wondered if the mutation frequency (the number of mutations across all patients) differs between histone genes encoding the same protein. Testing this possibility would reveal if histone gene mutation frequencies are primarily dependent on the encoded protein or gene. For this analysis, nine histone proteins that are encoded by multiple genes were examined (Figure 1C), and missense, silent, nonsense, nonstop, frameshift insertion, frameshift deletion, splice site, splice region, and translational start site mutations were considered. Within each cancer type, a Pearson’s Chi-Squared homogeneity test was performed to determine if the number of mutations differs between genes encoding the same histone protein (see Section 2). The patient mutation frequency among genes encoding H4 (Uniprot Accession: P62805), H3.1 (P68431), H3.2 (Q71DI3), and Histone H2B type 1-C/E/F/G/I (P62807) proteins were all found to differ significantly within the pan-cancer data set (q < 0.05, Benjamini–Hochberg, FDR = 0.05). Similarly, genes encoding H4, H3.1, and H3.2 proteins were differentially mutated in Squamous Cell Carcinoma, Endometrioid Adenocarcinoma and Infiltrating Duct Carcinoma, and Squamous Cell Carcinoma, respectively (q < 0.05), while no significant differences between genes were found for any other histone proteins in any other cancer types. Collectively, these results suggest that for some cancer types, gene-level factors rather than the protein sequence shape mutational frequency. These findings are perhaps not surprising given that mutagenic events function at the level of the DNA and are directly influenced by genomic factors.

### 3.5. Identification of Novel Histone Gene Drivers across Various Cancers

To identify novel driver histone genes across 68 cancer types, we applied the most recent version of MutSig2CV (v3.11). For our analysis, we used a stringent criterion (q-value ≤ 0.1) to determine driver histone genes in each cancer type. Our analysis identified seven histone genes as drivers across multiple cancer types (Table 1). Three drivers (*H2AC16*, *H1-8*, and *H1-5*) were found in squamous cell carcinoma, two drivers (*H4C4* and *H3C3*) in infiltrating duct carcinoma, one driver (*H1-2*) in hepatocellular carcinoma, one driver (*H2AC16*) in adenocarcinoma, and one driver (*H1-4*) in multiple myeloma (Appendix A). We note that our analysis did not find *H3-3A* and *H3-3B* as drivers of cancer despite both genes being known to harbor mutations that can drive pediatric glioblastoma [38]. We reason that our inability to capture these known histone gene cancer drivers likely reflects the limited number of pediatric samples in our data set (30 patients or 0.23% of the pan-cancer data set). Of the histone genes that emerged from our analysis, *H2AC16* was identified as a driver in two cancer types: squamous cell carcinoma and adenocarcinoma. Notably, of these seven histone driver genes, four (*H1-2*, *H1-4*, *H1-5*, and *H1-8*) encode a linker histone and three (*H2AC16*, *H4C4*, and *H3C3*) are core histone genes. Furthermore, six of these genes (*H1-2*, *H1-4*, *H1-5*, *H2AC16*, *H3C3*, and *H4C4*) encode canonical histones while only one (*H1-8)* encodes a variant (Appendix A). Interestingly, these novel potential driver genes exhibited intermediate mutation rates (Appendix A). Collectively, these findings emphasize the importance of going beyond mutational frequency to identify driver genes.

As it has been demonstrated that mutation rate in general is shaped by genomic and epigenomic factors [25], we examined how large-scale confounding factors (covariates) relate to the mutation rate in histone genes. Specifically, we examined if the mutation rate of histone genes is correlated to the replication time, chromatin state, expression level, and chromosomal location of histone genes. Overall, we found that the observed histone gene mutation rate is statistically significant and positively correlated with the replication time (Pearson correlation coefficient, PCC = 0.45, *p*-value < 0.01) and negatively correlated with the expression level (PCC = −0.38, *p*-value, 0.01). No significant linear correlation was detected with the chromatin state (Figure 3A–C).

Because our analysis primarily classified canonical histones as driver genes, we wondered if canonical and variant histones differed in the factors described above. Overall, we observed a separation of variant histone genes from canonical histone genes in terms of replication time, expression level, and chromatin state (Appendix A). Replication time refers to the relative time of the process of DNA replication during the S-phase of the cell cycle. Replication can be initiated at many genomic locations and follows a certain temporal order. Usually, active chromatin regions are open and are replicated earlier in the S-phase as compared to the repressed heterochromatin regions. We found that canonical histone genes were predominantly clustered in the intermediate replicating time regions of a more open chromatin state compared to all other genes (Figure 3A–C and Appendix A), whereas variant histone genes had earlier replication time. Among the predicted histone gene drivers, all six canonical histones (*H1-2*, *H1-4*, *H1-5*, *H2AC16*, *H3C3*, and *H4C4*) are located on chromosome 6, while the one variant histone (*H1-8)* is located on chromosome 3 (Appendix A). Compared to other genes, histone genes exhibited narrower distributions of replication time and expression levels, suggesting their tight regulation (Figure 3D–E). By contrast, the distribution of chromatin state values was wider for histone genes than for other genes (Figure 3F), although this result pertains only to a specific cell type, as discussed later.

### 3.6. Mutagenic Processes Shaping the Mutational Landscape of Histone Genes

To gain insight into the mutational processes that shape the mutational landscape of histone genes, the pan-cancer mutational data were divided into three groups: mutated histone genes (gene_group1), mutated non-histone genes (gene_group2), and genes mutated in patients without histone gene mutations (gene_group3) (Figure 2A) (see Section 2). MutaGene was used to obtain mutational profiles for each group, which were then decomposed using a predefined set of mutational signatures [32,39].

First, we observed similar mutational patterns dominated by C > T and C > A mutations in various trinucleotide contexts for all three gene groups (Appendix A). Nevertheless, the mutated histone genes displayed an increase in the number of C > G mutations in TCG, GCT, and TCT contexts and a decrease in the number of C > A mutations in the TCT context compared to gene_group2. This difference is particularly striking when one considers that gene_group1 and gene_group2 were derived from the same set of patients and cancer types, indicating that the differences cannot be attributed to the difference in the etiology of mutational processes operating in different patients or cancer types.

Second, the mutational signature analysis (Figure 2B) revealed five major signatures operating in gene_group1 (consisting of mutated histone genes). Among these signatures, the signatures associated with exogenous environmental factors, such as ultraviolet (UV) light (8.9% exposure) and aflatoxin (5.6% exposure), were observed. Signatures associated with endogenous mutational processes were also found, including APOBEC (apolipoprotein B mRNA editing enzyme) cytidine deaminases (9.6% exposure), defective DNA mismatch repair (MMR deficiency) (26.7% exposure), and defective DNA Polymerase Epsilon (POLE deficiency) (13% exposure). Similar to gene_group1, the UV-light signature, the APOBEC signature (5.8% exposure), MMR deficiency signature (24.7% exposure), and POLE deficiency signature (19.6% exposure) were also identified in gene_group2 (mutated non-histone genes from the same set of patients) (Figure 2C). However, certain mutational processes differed between these two groups. Particularly, the aflatoxin signatures were present in the mutated histone gene group but absent from the others, and the spontaneous deamination of 5-methylcytosine (clock-like signature) was present in gene_group2 but absent in gene_group1. The mutational landscape in gene_group3 (consisting of other non-histone mutated genes) was mainly attributed to UV-light and MMR-associated mutational processes, which contributed 15.9% and 15.8% of exposure, respectively (Figure 2D).

The mutational signature analysis showed that signatures associated with defective DNA Polymerase Epsilon were present only in patients harboring histone mutations (patient_set1). We examined the potential relationship between the presence of histone mutations and mutations in *POLE* and found that the presence of *POLE* mutations in cancer patients is associated with the presence of histone mutations (Fischer’s exact test one-sided *p*-value < 0.001, odds ratio of 6.01). Importantly, we found an even stronger association if known *POLE* driver mutations were considered (the total number of samples containing *POLE* driver mutations was 50, *p*-value < 0.001, odds ratio of 37.39). Overall, these findings point to a relationship between histone gene mutations and aflatoxin exposure or *POLE* mutations.

## 4. Discussion

Histones and their modifications play essential roles in epigenetic signaling, a process that is, in part, cell-type specific. As such, mutations in histone genes may vary in their effects across tissues and cancer types. Consistently, we observed that renal cell carcinoma, spindle cell melanoma, and undifferentiated sarcoma had a depletion of mutations in histone genes compared to other genes. By contrast, histone genes showed higher mutation rates in infiltrating duct carcinoma, clear cell adenocarcinoma, transitional cell carcinoma, and squamous cell carcinoma compared to other genes. These findings point to the possible association of histone gene mutations with sarcomas and cancers of epithelial tissue. Indeed, a previous study detected a significant amplification of chromosome 6 (harboring the largest histone gene cluster) in carcinosarcomas. Moreover, it showed that the expression of mutant H2A and H2B histones was associated with increased expression of markers of epithelial-mesenchymal transition [40].

The mutational frequency observed across patients depends on a wide range of factors, including the background mutation rate (mutability) of a particular nucleotide or a genomic region, replication time, histone modifications, gene expression, and chromatin state. Collectively, these factors have been shown to explain up to 86% of the variance in cancer mutation rates [41]. In line with this idea, we found that the observed mutation rate in histone genes had a significant positive correlation with replication time, as well as a weaker negative correlation with gene expression levels. No significant correlation was detected between histone gene mutation rate and chromatin state. However, a limitation of this analysis is that the Hi-C data used were generated in a single cell line (K562) and may not accurately reflect the chromatin states of the relevant cancer types examined, given that chromatin state has been reported to be, in part, cell-type specific [29].

While our approach to evaluate cancer driver status for histone genes aimed to control for genome-level factors, we note that the paucity of data sets prevents us from fully ruling out or capturing the breadth of cell type specific variability inherent in these factors. Nevertheless, replication time was reported previously for HeLa cells and shown to be highly similar across cell types and closely related species; evidence of gene expression conservation for some histone genes was previously reported [42,43].

Leveraging a large amount of mutational data, we identified seven previously unreported driver histone genes across various cancer types. Most genes belong to the linker histone H1 family. Consistently, recent studies identified a number of cancer mutations in H1 histone genes, especially in B cell lymphomas [44]. The key role of H1 in chromatin compaction raises the possibility that these mutations may lead to profound architectural remodeling of the genome [36,45]. Our study also identified driver genes in other histone families: H2A (*H2AC16*), H3 (*H3C3*), and H4 (*H4C4*). Previous studies have shown that histone driver mutations in core histones can disrupt various aspects of chromatin structure and function, affecting histone–DNA contacts, octamer stability, histone tail dynamics, nucleosome compaction, and the levels of histone post-translational modifications [46,47,48,49]. Whether mutations in our driver genes alter these or other cellular processes remains to be determined. Interestingly, the *H4C4* gene, which belongs to the highly evolutionarily conserved H4 family, was predicted as a driver in infiltrating duct carcinoma. Although our analysis failed to capture histone genes previously annotated as drivers (*H3-3A*, *H3-3B*, *H3C2*, and *H4C9*) and included in the CGC data set, we note that this is likely a result of a lack of pediatric samples in our data set, samples where these drivers are most common.

The high demand for canonical replication-dependent histones during the S-phase can be achieved by increasing the efficiency of their transcription and/or by increasing the number of genes that code for histone proteins. The former is accomplished by histone gene clustering on chromosomes to form the histone locus body, where histone mRNA biosynthesis factors are found in high concentration to make transcription more efficient. As we report, nine histone proteins are encoded by multiple histone genes. Four of these (H4, H3.1, H3.2, and Histone H2B type 1-C/E/F/G/I proteins) had significantly different mutation frequencies between their encoding genes within our pan-cancer data set and the squamous cell carcinoma and endometrioid adenocarcinoma data set. Consistently, even though there are eleven histone genes encoding the H4 histone protein, only one gene has been predicted as driver, as we discussed in the previous section. Collectively, these findings are consistent with gene-level factors (replication timings and gene expression) shaping the histone mutation rate of histones. In addition, our study suggests that all histone gene families are affected by mutations at similar rates, indicating a need to move beyond histone H3 and H1 to understand how histone mutations contribute to the oncogenic process.

Why mutations distribute unevenly across histone genes even though they encode the same histone type or protein remains enigmatic. In fact, genes encoding the same histone protein belong to gene clusters, localize to similar genomic regions, and have a very similar gene expression, replication time, and chromatin state, meaning that they should have similar background mutation rates. One possibility is that the differences in mutation rate between histone genes that code for the same protein emerge from differences in codon usage. While a relationship between codon bias and mutational frequency for histone genes remains to be examined, differences in codon usage across histone genes may exist as a way of limiting the loss of histone gene copies via homologous recombination between histone genes. Indeed, according to an elegant study modeling gene redundancy across evolution, the mutation rates of redundant genes are not necessarily similar. In fact, if a redundant gene with higher efficacy has a lower mutation rate than a gene with reduced efficacy, this can lead to equilibrium, with both copies of the genes maintained in the population [50].

Beyond identifying novel driver genes, our study identified specific mutational processes that affect histone genes. While this may be a specific property of histones, it may also reflect their genomic localization. Consistent with the latter, chromosome 6 (containing 60 histone genes), 1, 11 and 17 have been shown to be enriched with point mutations and large structural rearrangement compared to other chromosomes in radiation-induced tumors [51]. The underpinning of this effect is likely complex and might include the specific DNA context, codon usage, and others.

## 5. Conclusions

In this paper, we aimed to understand the complex relationship between the presence of certain mutational processes and other factors important for shaping the cancer mutational patterns in all human histone genes. Our results allow us to highlight the advantages and limitations of using canonical computational methods for histone driver predictions, pointing to the pressing need to develop a histone-specific analysis framework to gain better insight into the interplay between genetic and epigenetic factors in cancers that involve mutation to histone genes.

## Figures and Tables

**Figure 1 cancers-15-03437-f001:**
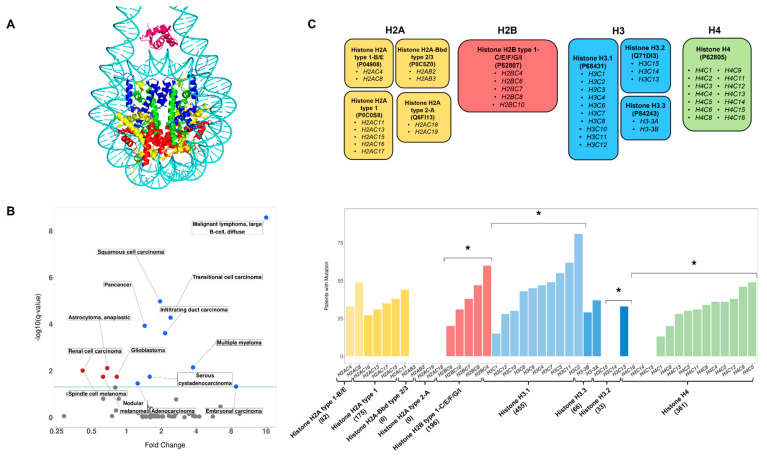
The Mutation Rate of Histone Genes in Cancer. (**A**): A depiction of the chromatosome derived from PDB structure 7K5X [1]. H1 (magenta), H2A (yellow), H2B (red), H3 (blue) and H4 (green) histones are color-coded based on histone type. DNA is colored cyan. (**B**): Volcano plot displaying the statistical significance (q-values) versus the fold change in mean mutation rate for histone proteins versus all non-histone proteins, within an individual cancer type. Fold change is calculated as (mean histone mutation rate)/(mean non-histone mutation rate). q-values are plotted on the y-axis on a −log_10_ scale (Mann–Whitney U, Benjamini–Hochberg, FDR = 0.05). The green line corresponds to q = 0.05. Labels are provided for cancer types with a significant difference between the mutation rates of histone proteins and non-histone proteins (q < 0.05). Furthermore, points with fold change > 1 and q < 0.05 are colored blue. Points with fold change < 1 and q < 0.05 are colored red. Points with q < 0.05 are colored in grey. (**C**): (**top**) Histone proteins encoded by multiple genes. A box represents a unique histone protein and contains information about the Uniprot accession number (in parentheses) and the genes that encode that protein. Boxes are colored to represent histone type. (**C**): (**bottom**) The number of pan-cancer patients with a mutation in each histone gene is shown in a bar plot. Histone genes are grouped by the protein they encode. Histone genes are been color-coded to indicate which histone family they belong to (yellow: H2A; red: H2B; blue: H3; green: H4). Shading is used to distinguish different histone proteins belonging to the same family in the bar plot. The total number of mutations per protein is displayed in parentheses below the protein name. Asterisks indicate instances where a Kruskal–Wallis test shows that mutation rates across genes encoding a common protein vary significantly (q < 0.05, Benjamini–Hochberg, FDR = 0.05). Bar plots are color-coded based on the histone family they represent.

**Figure 2 cancers-15-03437-f002:**
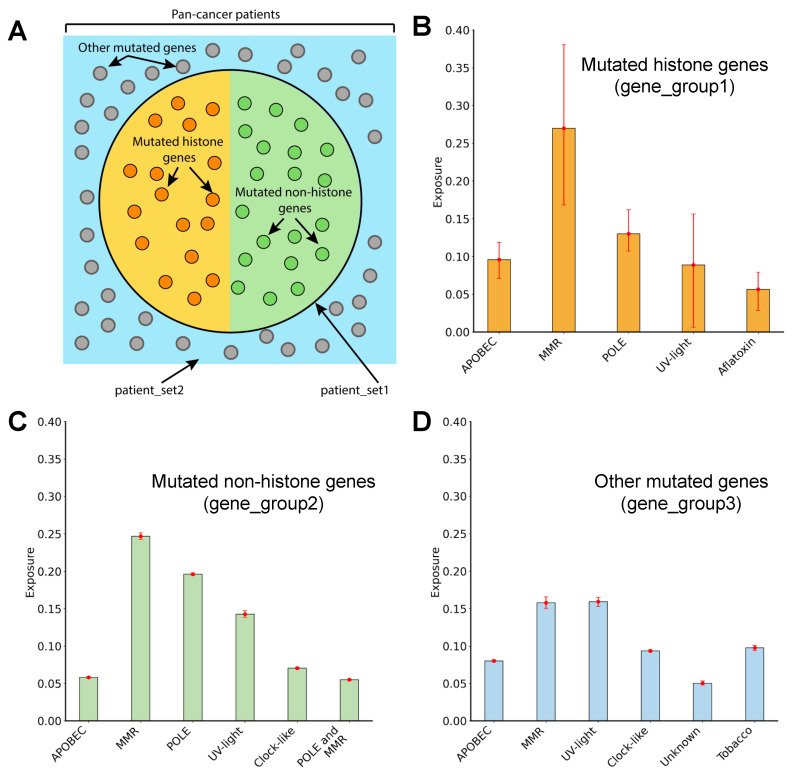
Mutational processes operating in histone and other genes. (**A**): A schematic representation of patient and gene groups used in the study from the pan-cancer mutation data. The large circle contains patient_set1 and the remainder represents patient_set2. Each small circle refers to a mutated gene within a given patient group: orange and green circles represent mutated histone genes (gene_group1) and mutated non-histone genes (gene_group2), respectively. Grey circles represent all mutated genes from patient_set2. (**B**–**D**) Estimated mutational signature exposure values for mutated histone genes, mutated non-histone genes, and mutated genes in patients without histone mutations. MMR refers to mutational signatures associated with a deficiency in DNA mismatch-repair. The POLE signature is related to defective Polymerase Epsilon. The Tobacco signature is linked to tobacco smoking. The Clock-Like signature is associated with spontaneous deamination of 5-methylcytosine. The POLE and MMR signatures refer to concurrent Polymerase Epsilon mutation and defective DNA mismatch repair. Aflatoxin refers to a signature related to aflatoxin exposure. Unknown refers to a signature of unknown etiology. Exposures of signatures with common etiologies were combined. Error bars represent 95% bootstrap confidence intervals (see Section 2).

**Figure 3 cancers-15-03437-f003:**
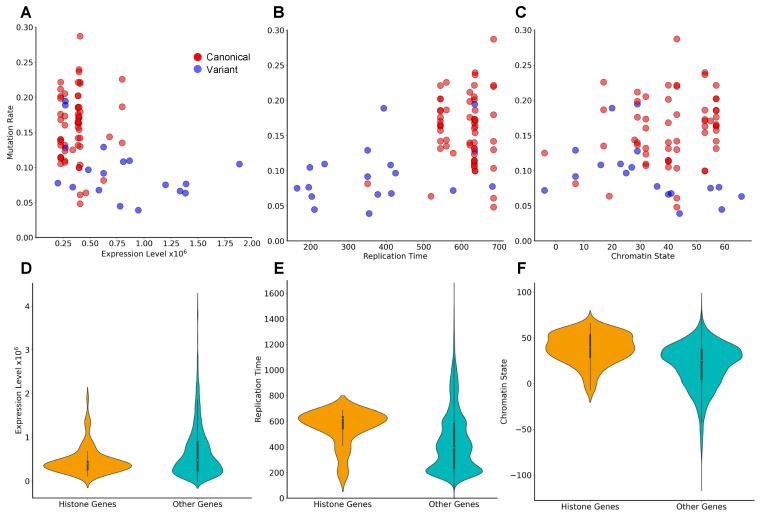
Histone gene mutation rate is associated with histone gene expression level and replication time but not its chromatin state. Scatterplots showing the pan-cancer mutation rate vs. expression level (**A**), replication time (**B**), and chromatin state (**C**) for all canonical (red circles) and variant (blue circles) histones. Mutation rate is associated with expression level and replication time. (**D**–**F**) The distribution gene expression (**D**), replication time (**E**), and chromatin state (**F**) values for histone genes compared to genes coding non-histone proteins. Histone genes show a wider range of gene expression, replication time, and chromatin state values compared to gene coding non-histone proteins.

**Table 1 cancers-15-03437-t001:** Novel histone driver genes identified in various human cancers.

Gene Symbol	Protein	Type	Chromosome Location	q-Value	Cancer Type
H1-4	H1.4 linker histone, cluster member	Canonical	Chr6	0.002	Multiple myeloma
H2AC16	H2A clustered histone 16	Canonical	Chr7	0.004	Squamous cell carcinoma
H1-8	H1.8 linker histone	Variant	Chr3	0.006	Squamous cell carcinoma
H1-5	H1.5 linker histone, cluster member	Canonical	Chr6	0.009	Squamous cell carcinoma
H2AC16	H2A clustered histone 16	Canonical	Chr7	0.049	Adenocarcinoma
H4C4	H4 clustered histone 4	Canonical	Chr6	0.062	Infiltrating duct carcinoma
H3C3	H3 clustered histone 3	Canonical	Chr6	0.072	Infiltrating duct carcinoma
H1-2	H1.2 linker histone, cluster member	Canonical	Chr6	0.099	Hepatocellular carcinoma

## Data Availability

The TCGA, MMRF-CoMMpass, HCMI, CMI, and CPTAC somatic mutation data are available for download from the GDC Data Portal repository (https://portal.gdc.cancer.gov [portal.gdc.cancer.gov]). Known driver genes are available for download from the COSMIC-CGC (https://cancer.sanger.ac.uk/census). Coding sequences are available for download from the NCBI-CCDS (https://www.ncbi.nlm.nih.gov/projects/CCDS/CcdsBrowse.cgi).

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
