# Peer review of "Leveraging Gene Redundancy to Find New Histone Drivers in Cancer"

_cancers, 2023, doi:10.3390/cancers15133437_

Round 1

Reviewer 1 Report

This work discusses an important aspect of the role of histone mutagenesis in emerging new cancer drivers. Specifically, a large dataset of histone gene mutations in cancer was built to investigate mutations in 96 human histone genes associated with 68 different cancer types. Authors found that several types of cancers are outstanding in terms of the nuber of mutations in histone genes – either revealing increased or decreased numbers of those. Authors pay specific attention to the phenomenology of canonical and variant histones. Overall, this work present an important view on potential ways emergence of cancer drivers in histone genes, and it should be published as is. Several questions below could be addressed by authors to further clarify their fundings and conclusions. 

Figure 2. Please, provide an explanation how Hi-C data is used to characterize teh difference between the histone and other genes (charts C and F).

It would be interesting to see authors’ opinion why most of the cancer drivers were found in canonical histones from the evolutionary perspective in addition to very interesting discussion “Why mutations distribute unevenly across histone genes”.

Discussion on prevalence of C->G in histone genes could be of interest. Please, provide if possible.

Discussion emphasizes on the key role of H1 histone in chromatin structure compaction, which may hint on the corresponding mechanism of the cancer drivers in H1 histine genes. What about mechanisms of cancer drivers in other histones? Do they work by affecting the nucleosome structure-folding or contacts with DNA? Please, provide your opinion on that.

Reviewer 2 Report

The authors examine various aspects of histone gene mutation in a large compendium of cancer datasets derived from exome sequencing projects.  The paper is well written and the results presented clearly.  The conclusions cited in the abstract are supported by the evidence, and as far as I can tell the methods used are up to date and correctly carried out.  The central idea is a clever use of redundancy among histone genes to search for mutational bias, driver genes etc.  Overall, I found the manuscript to be interesting and of course relevant to cancer researchers and clinicians.  As far as I can tell the work is novel, which is a little surprising considering the potential importance of histone genes in cancer.

I have only small suggestions for improvement.  Figure 1B is a little hard to read - could that be improved?

In a few cases, the results are described a little bluntly, with no attempt to explain them.  There are a few surprising results that might merit a little more discussion.  In particular, no evidence for differences in mutation rates across the histone families was found, a little surprising.  Could there be some underlying reason for this, or are there limits to the sensitivity addressable by the dataset?  Similarly, no evidence for epigenetic status on mutation rate.  Is this not also surprising?  I'm not questioning the author's findings, but I was curious if they feel this is a real (negative) finding or if there are inherent limits in the data.

Up to the authors as to whether they want to address some of these questions - overall the paper is well written and quite succinct, and I would recommend publication.
